# HDSI: High dimensional selection with interactions algorithm on feature selection and testing

**Rahi Jain[1], Wei Xu[1,2]** *

**1** Biostatistics Department, Princess Margaret Cancer Research Centre, Toronto, Ontario, Canada, **2** Dalla Lana School of Public Health, University of Toronto, Toronto, Ontario, Canada

* wei.xu@uhnresearch.ca

## Abstract

Feature selection on high dimensional data along with the interaction effects is a critical challenge for classical statistical learning techniques. Existing feature selection algorithms such as random LASSO leverages LASSO capability to handle high dimensional data. However, the technique has two main limitations, namely the inability to consider interaction terms and the lack of a statistical test for determining the significance of selected features. This study proposes a High Dimensional Selection with Interactions (HDSI) algorithm, a new feature selection method, which can handle high-dimensional data, incorporate interaction terms, provide the statistical inferences of selected features and leverage the capability of existing classical statistical techniques. The method allows the application of any statistical technique like LASSO and subset selection on multiple bootstrapped samples; each contains randomly selected features. Each bootstrap data incorporates interaction terms for the randomly sampled features. The selected features from each model are pooled and their statistical significance is determined. The selected statistically significant features are used as the final output of the approach, whose final coefficients are estimated using appropriate statistical techniques. The performance of HDSI is evaluated using both simulated data and real studies. In general, HDSI outperforms the commonly used algorithms such as LASSO, subset selection, adaptive LASSO, random LASSO and group LASSO.

## Introduction

Classical statistical models have been the mainstay for data analysis. However, the growth in dataset sizes both in terms of sample size ($n$) and feature dimension ($p$) had triggered some challenges for traditional approaches of statistical data analysis. In the case of $p>n$, classical approaches cannot control for false discovery rate of identified features. In the case of $n>p$ sample size, an increase in the size of a feature set would exponentially increase the feature combination set that needs evaluation which leads to an exponential increase in requirement for computation time and resources [1]. Furthermore, the original input feature set might not contain complete information. Hence, the incorporation of interaction terms in the feature set

**Data Availability Statement:** All real study dataset files are available from the ICPSR database (accession number(s) Dataset I: https://healthdata. gov/dataset/community-health-status-indicators-chsi-combat-obesity-heart-disease-and-cancer,

Dataset II: https://www.icpsr.umich.edu/web/
NACDA/studies/36873, Dataset III: https://www.
icpsr.umich.edu/icpsrweb/NACDA/studies/32961,
Dataset IV: https://www.icpsr.umich.edu/icpsrweb/
NACDA/studies/20541, Dataset V: https://www.
icpsr.umich.edu/icpsrweb/NAHDAP/studies/
36389).

**Funding:** The authors received the following
funding support: Canadian Network for Research
and Innovation in Machining Technology, Natural
Sciences and Engineering Research Council of
Canada, RGPIN-2017-06672, Dr Wei Xu; Prostate
Cancer Canada, Dr Wei Xu.

**Competing interests:** The authors have declared
that no competing interests exist.

might be necessary [2], which exacerbates the challenge of processing the feature combination set.

Feature selection has been used as a common approach to address the high dimensionality challenge by identifying the significant features from the input feature space for performing final statistical analysis. Different strategies are in use for performing feature selection which can be categorized into three primary types and two sub-types, as shown in Fig 1. The first primary approach is expert-based feature selection which relies upon the use of experience of experts to shortlist variables for final model development [3, 4]. It is subjective and becomes more challenging to perform with an increase in feature set and interaction terms.

Statistics-based feature selection is another primary approach which uses various statistical techniques to perform the feature selection. One strategy is to select features based on the intrinsic properties of the features like multicollinearity [5] and distribution [6]. Another strategy is to select features based on their statistical significance (i.e. p-value) during univariate analysis [4, 5]. Linear regression-based screening is a common approach in univariate analysis [7]. The third strategy is to select features based on their importance in the model during multivariate analysis [4, 5]. Some of the conventional approaches used in multivariate analysis are subset selection and penalized regression [2, 8]. However, they have certain limitations like subset selection approach can work only if $p<<n$ and LASSO regression cannot select features more than $n$ [9]. Besides classic LASSO, many extensions have been proposed. Elastic LASSO can be used for $p>n$ cases, however, it may not be immune from selecting the noise variables [10]. Group LASSO is another approach which has been used to improve the performance of LASSO in selecting variables which need to be selected in a group such as marginal and interaction terms. However, they follow a selection hierarchy and may not select interaction terms if marginal features are not selected. Further, the groups need to be predefined [9].

One of the sub-type approaches is expert-statistics hybrid based feature selection. This approach incorporates the domain knowledge in the feature selection process. The domain knowledge and statistical analysis sequence can occur in two ways. One strategy is to first assign importance to the features based on the domain knowledge followed by implementation of statistical analysis on the feature set for feature selection. This approach is inherent in Bayesian regression-based approaches [11]. The second strategy is to perform statistical analysis for preliminary feature selection. The final features are selected from the preliminary selected feature set based on their importance estimated from the domain knowledge. Differential gene

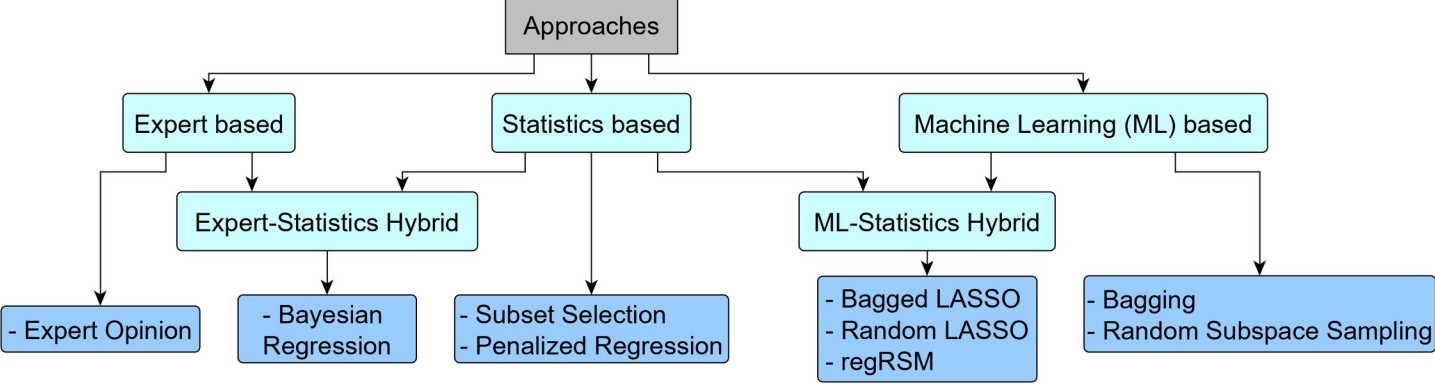

**Fig 1. Feature selection approaches suggested in literature with examples.**

expression analysis commonly employs this strategy [12]. However, the issues of expert-based feature selection approach prevail.

The third primary approach is machine learning (ML) based feature selection which can deal with high dimensional data but focus less on interpretability of the selection process. Random Forest is a common ML technique which can perform feature selection [13]. It provides the importance of a feature in model prediction, which can be a metric for feature selection. The technique allows the use of different ensemble methods like bagging [14], random subspace sampling [15] or both [16] for model building. Bagging and random subspace sampling methods enable creating robust models in noisy settings [17].

Consequently, the ML ensemble methods have been integrated with classical statistical techniques to create ML-statistics hybrid based feature selection as the other sub-type approach. BoLASSO is one such technique which performs LASSO regression on different bootstrap samples of $n$ [18]. The features are selected based on the number of models in which feature was selected. regRSM is another technique in which subsamples of feature set are created with/without weighted selection probability followed with linear regression on each of the feature subsets [19]. The feature selection is based on their performance in t-statistic metric across different models. Random LASSO technique is a two-step procedure which integrates both bagging and random subspace sampling with LASSO for feature selection [20]. Bootstrapping and random subspace sampling creates subsamples of a dataset. In the first step, application of LASSO on each subsample provides the importance of each feature. In the second step, application of LASSO or adaptive LASSO on each subsample enables in obtaining the final coefficient estimate of the features.

All these techniques do not directly consider the interaction effects in the feature selection process. In the case of LASSO-based techniques, they may select only the interaction feature while ignoring their marginal features. Further, random LASSO is comprehensive in feature selection and could outperform elastic net [20], but it is also computationally intensive due to its two-step approach. Random LASSO performs both feature selection and estimation but does not give the user flexibility to stop the process at feature selection or use techniques other than LASSO or adaptive LASSO. It does not provide a guideline to address the interaction terms in the model as well as the significance of estimated coefficients. Further, the estimates may suffer from systematic bias as during feature selection and estimation it assigns zero value to features not sampled. Additionally, its selection step may have interpretability issues as it uses a user assigned threshold value to determine the feature performance.

In this paper, we focus on addressing the challenge of incorporating interaction effects in the feature selection approach with lesser computational intensiveness as compared to random LASSO. We propose a novel strategy, named High Dimensional Selection with Interactions (HDSI) algorithm for improved feature selection. Our approach combines bootstrapping and random subspace sampling with classical statistical model selection techniques with the in-built capability to handle interaction terms. Further, the approach addresses the shortcomings of random LASSO by allowing use of multiple statistical techniques in feature selection, reduce the systematic bias in feature selection and improve interpretability of feature selection process. The paper is organized as follows. The proposed strategy is explained in the Methodology Section with the evaluation of the method performance in the Simulation Study Section. The strategy is tested using real data studies in the Real Data Studies Section, followed by the Conclusion and Discussion Section.

## Methodology

In this section, first, we will describe some of the existing feature selection algorithms such as LASSO, random forest and random LASSO and is followed by the proposed HDSI algorithm.

### LASSO

LASSO is a penalized regression method proposed by (Tibshirani, 1996) [21] which enables the features selection. Its criterion performs L1-penalization of the regression coefficients, $\min_{\beta} \sum_{i=1}^{n} (y_i - \sum_{j=1}^{p} \beta_j x_{ij})^2 + \lambda \sum_{j=1}^{p} |\beta_j|$, where $i$ represents the $i^{th}$ subject of totally $n$ subjects, $\beta_j$ represents the regression coefficient of $j^{th}$ feature in $p$ feature set. $y_i$ represents the response feature and $x_i = (x_{i1}, \ldots, x_{ip})$ represents a $p$-dimensional vector of features. $\lambda$ is a non-negative tuning parameter.

The L1-norm penalty has the singularity of the derivative at $|\beta_j| = 0$, so LASSO tries to shrink the $\beta_j$ towards zero and some estimated $\beta_j$ will be precisely zero at sufficiently large $\lambda$. However, this method has a few limitations. Firstly, in cases of $p>n$, it can only select maximum $n$ features. Secondly, among the highly correlated variables, it may choose arbitrarily only one variable and drop the other variables [10]. Thirdly, in case of interactions, these methods could select the interaction terms while dropping the main features, since LASSO is neutral to the pattern in which features exist [22]. In cases where interpretive models are desired, methods that allow retention of main features is preferable.

### Random forest

Random Forest is the extension of the decision tree method (a non-parametric approach) [16]. In the decision-tree technique, variables selection is made parsimoniously based on a series of logical criteria to separate the data into subsets and estimate the outcome in each subset. The random forest creates multiple decision-trees. These multiple decision-trees could be prepared by either bootstrapping the samples, randomly selecting the features or both. The random forest estimates the overall importance of features based on the influence of feature on the performance of multiple decision-trees using some metric [16, 23, 24]. The features can be selected using the feature importance scores. However, the approach may show bias due to the scale of measurement of features [24]. Further, while, the random forest consider interaction terms in building the model, it does not directly provide the importance score of the interaction effects [25].

### Random LASSO

Random LASSO is a dual procedure method as shown in Algorithm 1 [20]. In the first procedure, the importance of features is estimated using bootstrapping and random subspace sampling. LASSO is used for feature estimation in each bootstrap. Then, in the second procedure, bootstrapping and weighted random subspace sampling is performed to get the estimates of the coefficients of features. LASSO (or, Adaptive LASSO) is used for feature estimation in each bootstrap. The final estimates of the features are computed by averaging multiple estimates obtained from bootstrapping over the total number of bootstraps.

| Algorithm 1: Random LASSO | |
|---|---|
| Procedure I | Generate the Importance Scores for the Features |
| I(a) | Bootstrap $B$ samples with size $n$ from the original dataset. |
| I(b) | For each Bootstrap sample, randomly select $q^{(1)}$ features ($q^{(1)} \leq n$) from original $p$ features. |
| I(c) | Apply LASSO to estimate coefficient, $\widehat{\beta}_{ij}^{(1)} \vert i = \{1, \ldots, B\}, j = \{1, \ldots, p\}$. The coefficients of unselected features for each bootstrap sample are considered zero. |
| I(d) | Compute the importance score, $I_j = \sum_{i=1}^{B} \vert \widehat{\beta}_{ij}^{(1)} \vert / B$. |
| Procedure II | Generate the final coefficient estimates of the Features |
| II(a) | Bootstrap another set of $B$ samples with size $n$ from the original dataset |
| II(b) | For each Bootstrap sample, randomly select $q^{(2)}$ features ($q^{(2)} \leq n$) from original $p$ features with feature selection probability proportional to importance scores, $I_j$. |
| II(c) | Apply LASSO (or, Adaptive LASSO with weight, $w_j = I_j^{-1}$) to estimate coefficient, $\widehat{\beta}_{ij}^{(2)} \vert i = \{1, \ldots, B\}, j = \{1, \ldots, p\}$. The coefficients of unselected features for each bootstrap sample are considered zero. |
| II(d) | Compute the final coefficient estimate, $\widehat{\beta}_j = \sum_{i=1}^{B} \vert \widehat{\beta}_{ij}^{(2)} \vert / B$. The features with final coefficients above predefined threshold are selected. |

## High dimensional selection with interactions (HDSI)

Random LASSO is a comprehensive approach for feature selection in high dimensional settings, but it has certain limitations. One of the limitations is that it is computationally intensive and provides little flexibility in the coefficient estimation of selected features as it allows use of only LASSO (or, Adaptive LASSO) for coefficient estimation. Secondly, random LASSO does not consider any interaction terms in the feature selection process. Thirdly, random LASSO does not provide any statistical guideline for various activities like the number of bootstrap samples, feature selection and its significance estimation. Fourthly, random LASSO assigns zero value to coefficients of the features unselected during random selection. These values may create systematic bias as they are used to calculate the importance scores of features.

The HDSI methodology (Algorithm 2) is developed to address its limitations. Fig 2 provides a graphical representation of HDSI. The method generates random samples by bootstrapping the original dataset and random subspace sampling of the features. Interaction terms of sampled features are generated for each sample. The statistical modeling is performed on each sample using appropriate feature selection techniques like penalized regression or subset selection. The results of samples are pooled to determine the statistical significance of the estimated coefficients of features and select the significant features. The final coefficient estimation of the selected features could be done with another set of appropriate statistical modeling technique like simple linear regression. The proposed method is discussed below for more details.

**Procedure flexibility.** Random LASSO uses a two-stage procedure to perform feature selection and coefficient estimation. However, double bootstrapping makes the method computationally expensive. Besides that, while adaptive LASSO has *global* oracle property, its performance is dependent on the estimator used for assigning weights to the coefficients of predictors [26]. Further, random LASSO may bias the coefficients [20]. Hence, the user might prefer using other techniques on selected features for coefficient estimation, but the whole process needs to complete to get the feature selection results.

HDSI modifies random LASSO to increase procedure flexibility and reduce computation intensiveness. The first modification is using the importance scores from Step I(d) as a metric

for the feature selection. This modification allows the user to obtain preliminary results on feature selection and enable them to decide on the coefficient estimation step. The second modification is to allow the use of techniques other than LASSO for coefficient estimation in Procedure I. This will enable the user to use different techniques in the same procedure and address the LASSO limitations.

**Algorithm 2: HDSI**

| Procedure I | Feature Selection |
|---|---|
| I(a) | Bootstrap $B$ samples with size $n$ from the original dataset. |
| I(b) | For each Bootstrap sample, randomly select $q$ features ($q \leq n$) from original $p$ features. |
| I(c) | Prepare the $\chi$ interaction set from $q$ features. |
| I(d) | Create a final sample feature set, $p^* = \chi \cup q$. |
| I(e) | Apply LASSO (or, any other feature selection technique) to estimate coefficient, $\widehat{\beta}_{ij} \mid i = \{1, \ldots, B\}, j = \{1, .., p, .., \sum_{k=2}^{\omega} \binom{p}{k}\}$. The coefficients of unselected features for each bootstrap sample are considered missing. |
| I(f) | Compute the mean coefficient estimate, $\widehat{\beta}_j = \sum_{i=1}^{B} \widehat{\beta}_{ij}/b_j$, where $b_j$ is the number of bootstrap samples containing the $j^{th}$ feature.<br>Compute $j^{th}$ feature minimum coefficient of determination ($R^2$), $minR_j^2 = Min(R_{ij}^2) \mid i \in \{1, \ldots, B\}$. |
| I(g) | Select significant features based on the quantile of coefficient estimate and minimum $R^2$ value. |
| I(h) | Add missing marginal features of significant interaction terms in the final selected feature set. |

**Feature selection.** Random LASSO performs the feature selection in its first stage of the procedure. Similar to random LASSO (Step I(a) and Step I(b), Algorithm 1), HDSI (Step I(a) and Step I(b), Algorithm 2) generates random samples from the dataset of $n$ sample size and $p$ feature space. Multiple samples of size $n$ from the original dataset is created with a replacement through bootstrapping. For each of these samples, a feature sample of size $q$ from original feature space is created without replacement through random subspace sampling.

While, random LASSO performs LASSO based statistical modeling for coefficient estimation of each feature $q$ in each sample (Step I(c), Algorithm 1), HDSI incorporates interaction terms before statistical modeling (Step I(c) and Step I(d), Algorithm 2). HDSI generates all possible interaction terms between $q$ features. A final sample feature set of size $p^*$ is created for each bootstrap sample which is the combination of $q$ features and its interaction terms. The statistical modeling of $p^*$ features is performed (Step I(e), Algorithm 2). Different modeling techniques like LASSO, adaptive LASSO, regression and subset selection could be used for getting coefficient estimates of features in each of the bootstrap samples. HDSI uses three different techniques for feature selection, namely LASSO based (HDSI_L), adaptive LASSO based (HDSI_AL) and regression based (HDSI_R).

Random LASSO pools the coefficient estimates of all features from all the bootstraps to estimate the feature performance. It calculates the importance score of each feature by averaging its estimates from bootstrapped datasets over the total number of bootstraps (Step I(d), Algorithm 1). Random LASSO considers the coefficients of unselected features in a bootstrap sample as zero rather than missing, which causes systematic bias in the estimation of regression coefficients. HDSI estimates the feature performance using two metrics, namely coefficient estimates and model coefficient of determination ($R^2$). HDSI treats the coefficients and model

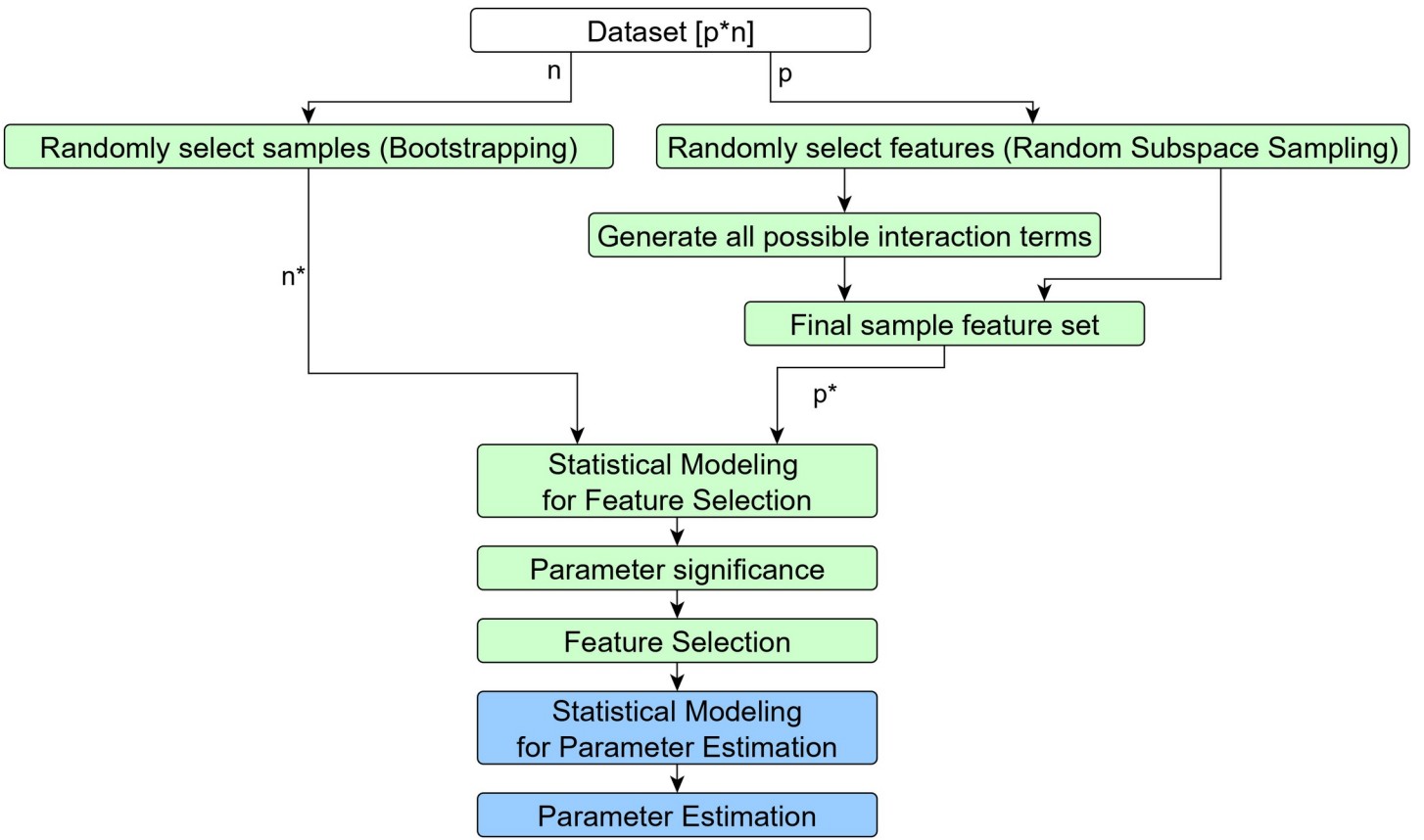

**Fig 2. Graphical representation of HDSI methodology.** p: number of features in the dataset, n: sample size of the dataset, p*: number of features (including sampled features and its interaction terms) used for modeling and n*: each bootstrapped dataset has the same sample size n used for modeling. The green colour steps represent the proposed method and blue colour steps represent the process to estimate the coefficient of selected features.

$R^2$ of unselected features in a bootstrap sample as missing (Step I(f), Algorithm 2). Consequently, during the computation step of the $j^{th}$ feature, bootstrap samples with the missing feature were dropped from the computation, as shown below:

$$\widehat{\beta}_j = \sum_{i=1}^{B} \widehat{\beta}_{ij} / b_j \qquad (1)$$

$$minR_j^2 = Min(R_{ij}^2)|i \in \{1, \ldots, B\} \qquad (2)$$

where $b_j$ is the number of bootstrap samples containing the $j^{th}$ feature, $\widehat{\beta}_j$ represents an averaged coefficient estimation of the $j^{th}$ feature over $b_j$ bootstrap samples and $minR_j^2$ represents a minimum model $R^2$ of the $j^{th}$ feature over $b_j$ bootstrap samples.

Random LASSO does not provide any statistical guideline to determine the significance of estimated coefficients. Since, coefficient estimate distribution is unknown, in HDSI the upper and lower quantile values of coefficients are estimated using $\widehat{\beta}_{ij}|i = 1, \ldots, b_j$ values as sample data. If zero value is not present between the lower and upper quantile values, $\widehat{\beta}_j$ is considered to have a significant non-zero value.

Finally, the features with performance above a predefined threshold are selected. Random LASSO uses heuristic rather than the statistical threshold to select the features based on the estimated coefficients (Step II(d), Algorithm 1). HDSI selects the features which fulfil two criteria (Step I(g), Algorithm 2). Firstly, the coefficient estimates between lower and upper quantile values should not contain zero. The quantile ($Q_i$) is a hyperparameter which needs to be optimized where lower quantile is $Q_i/2$, and upper quantile is $100-Q_i/2$. Secondly, the features should have occurred in models with a high coefficient of determination, i.e. $R^2$. A heuristic cut-off value ($R_f$) for considering high $R^2$ can be used. However, the heuristic cut-off value may not be easy to interpret. Accordingly, the heuristic cut-off value is transformed into $\mu_{minR^2} + R_f\sigma_{minR^2}$, where $\mu_{minR^2}(= \sum_{j=1}^{p^*} minR_j^2/p^*)$ is the mean value of $minR_j^2$ values of p* features. $\sigma_{Min_{R^2}}$ is the standard deviation of $minR_j^2$ values of p* features as given below.

$$\sigma_{minR^2} = \sqrt{\frac{\sum_{j=1}^{p^*}\left(minR_j^2 - \mu_{minR^2}\right)^2}{p^* - 1}} \tag{3}$$

$minR_j^2$ of the feature should be higher than $\mu_{minR^2} + R_f\sigma_{minR^2}$ for selection. The hyperparameter $R_f$ can take value from $[-\infty,\infty]$. Among the selected features, some interaction terms could get selected while their corresponding marginal features are not. In such cases, the final set of selected features incorporates the unselected marginal features of selected interaction terms (Step I(h), Algorithm 2).

**Interaction effects.** Random LASSO does not provide any guideline in dealing with interactions among the features. Further, non-group LASSO based LASSO methods have issues in dealing with interactions since LASSO is neutral to feature pattern; hence any pattern of features can exist [22]. Thus, a model based on LASSO and adaptive LASSO cannot deal with feature interactions. In HDSI, after the selection of $q$ features for a bootstrap sample, all the possible $k = \{2,\ldots,\omega\}$ level interaction terms, $\chi$, among the $q$ features are created. A new sample feature set, $p^*$ ($= \chi \cup q$) is used for coefficient estimation.

$$\chi = \bigcup_{k=2}^{\omega}\binom{q}{k} \tag{4}$$

**Number of feature samples.** The pooling criteria is dependent on the feature performance in different models. Accordingly, it is vital to ensure that every feature is sampled multiple times. The current random LASSO algorithm provides no statistical guideline to determine the number of bootstrap times a feature should be selected and modeled.

The HDSI method considers the hypothetical population mean value of a coefficient as zero, then uses it as a reference value against which the estimated mean value of coefficient could be compared. Since it is a one-sample case, the sample size of the coefficient values for a feature could be estimated from following Lehr's equation, $L = 8/\Delta2$ [27]. $L$ is the minimum number of times a feature should be selected (i.e., the sample size of coefficient values) and $\Delta$ is the effect size. As Cohen's rule of thumb, $\Delta$ could be equal to 0.2, 0.5 or 0.8 for 'small', 'medium' and 'large' effect sizes [28]. The probability of a feature to be included in a sample of $q$ features for any bootstrap is $\rho = q/p$. In cases when interaction terms are considered $\rho$ can be calculated as follows:

$$\rho = \sum_{k=2}^{\omega}\binom{q}{k}/\sum_{k=2}^{\omega}\binom{p}{k} \tag{5}$$

$q$ and $p$ are not added into the calculation because the marginal features would be selected when the interaction terms are selected. Each $B$ can be considered an independent trial for

selecting a feature. Then, the probability of a feature to get selected $L$ times, $(Pr(X = L))$, in $B$ trials is equivalent to the probability mass function of a binomial distribution, $\binom{B}{L} \rho^L (1 - \rho)^{B-L}$. Since, $L$ is the minimum number of desired selections of a feature, the cumulative distribution function should be used as follows for calculating $B$:

$$\Pr(X \geq L) = 1 - \sum_{m=0}^{L-1} \Pr(X = m) \tag{6}$$

$$B \geq f(\Pr(X \geq L), L) \tag{7}$$

**Feature estimation.** Random LASSO estimates the coefficient values of selected features by performing the second stage of the procedure (Algorithm 1). The second stage repeats the steps of the first stage of the procedure with some modifications. Firstly, weighted random subspace sampling rather than random subspace sampling performs the sampling of features. Secondly, adaptive LASSO (recommended) performs statistical modeling. In both modifications, the feature weight is proportional to its importance score obtained from step 1(d). The importance score obtained in Step 2(d) is the estimated coefficient value of the features. The features with the coefficient value above the heuristically determined threshold value are selected.

HDSI does not perform stage two to reduce computation time. Further, the statistical technique will also depend on the user problem statement. The method gives the user flexibility in choosing the appropriate statistical modeling technique for feature estimation. The current study uses ordinary least squares based regression.

**Hyperparameters.** HDSI requires three hyperparameters, namely, number of features in a sample ($q$), coefficient estimate quantile threshold ($Q_i$) and minimum $R^2$ threshold ($R_f$). The values of these hyperparameters depend on the dataset. Hence, hyperparameter optimization is needed for optimal performance. Hyperparameter optimization is done in three steps. In first step, $Q_i$ and $R_f$ are kept constant at value five and value zero respectively and $q$ is optimized. Randomly, multiple values are generated for $q$ and value with the best predictive performance is selected. In second step, $q$ and $R_f$ are kept constant at the best value obtained from first step and value zero respectively, while $Q_i$ is optimized. Randomly, multiple values are generated for $Q_i$ and value with the best predictive performance is selected. In third step, $q$ and $Q_i$ are kept constant at the best value obtained from step one and two respectively, while $R_f$ is optimized. Randomly, multiple values are generated for $R_f$ and value with the best predictive performance is selected.

**Simulation studies.** Simulated data are used to demonstrate the performance of the proposed method and compare it with other methods. The current study considers only two-way interactions, but the approach can be demonstrated for higher-order interactions too. The simulation data are generated from the regression model, $y = \beta_0 + \beta_1 x_1 + \cdots + \beta_p x_p + \beta_{12} x_{12} + \cdots + \epsilon$. $\varepsilon \sim N(0, \sigma^2)$, $x_1, \ldots, x_p \sim N(0, 1)$ and $\{x_{12}, x_{13}, \ldots, x_{(p-1)p}\}$ represents the two-way interactions between features $\{(x_1, x_2), (x_1, x_3), \ldots, (x_{p-1}, x_p)\}$. Coefficient values are zero for features unless mentioned (Table 1). Covariance matrix is defined to create multicollinearity in the model with non-zero covariance among $\{x_1, \ldots, x_5\}$ and zero covariance

**Table 1. Description of the simulation data.**

| Scenario | Features Effect | $\beta$ (Non-Zero coefficients) | p | Train set | Test set | $\sigma^2$ |
|---|---|---|---|---|---|---|
|  | *Marginal Terms* |  |  |  |  |  |
| 1 | Yes | $\{\beta_1, \beta_2, \beta_3, \beta_{12}\} = \{0.2, 0.3, 0.4, 0.3\}$ | 25 | 500 | 500 | 0.25 |
| 2 | Yes | $\{\beta_1, \beta_2, \beta_3, \beta_{12}\} = \{0.2, 0.3, 0.4, 0.3\}$ | 50 | 500 | 500 | 0.25 |
| 3 | Yes | $\{\beta_1, \beta_2, \beta_3, \beta_{12}\} = \{0.2, 0.3, 0.4, 0.3\}$ | 100 | 500 | 500 | 0.25 |
| 4 | No | $\{\beta_3, \beta_{12}\} = \{0.4, 0.3\}$ | 50 | 500 | 500 | 0.25 |

among all other cases as shown below:

$$
\begin{bmatrix}
x_1 x_1 & x_1 x_2 & . & . & x_1 x_5 & . & . & . \\
x_2 x_1 & x_2 x_2 & . & . & x_2 x_5 & . & . & . \\
\vdots & \vdots & \vdots & \vdots & \vdots & \vdots & \vdots & \vdots \\
x_5 x_1 & x_5 x_1 & . & . & x_5 x_5 & . & . & . \\
x_6 x_1 & x_6 x_1 & . & . & x_6 x_5 & . & . & . \\
. & . & . & . & . & . & . & . \\
x_p x_1 & . & . & . & . & . & . & .
\end{bmatrix}
=
\begin{bmatrix}
1 & 0.3 & 0.3 & 0.6 & 0.6 & 0 & . & 0 \\
0.3 & 1 & 0.3 & 0.2 & 0.1 & 0 & . & 0 \\
0.3 & 0.3 & 1 & 0.2 & 0.1 & 0 & . & 0 \\
0.6 & 0.2 & 0.2 & 1 & 0.1 & 0 & . & 0 \\
0.6 & 0.1 & 0.1 & 0.1 & 1 & 0 & . & 0 \\
0 & 0 & 0 & 0 & 0 & 1 & . & 0 \\
. & . & . & . & . & . & . & 0 \\
0 & 0 & 0 & 0 & 0 & 0 & 0 & 1
\end{bmatrix}
$$

The data for each feature is generated from the multivariate normal distribution. Table 1 shows the different settings considered for evaluating the different models. The number of features, $p$, considered across different settings varied from 25–100. The training and test dataset used in each scenario is 500. Scenario 4 does not consider the marginal effect of the interaction terms. The number of target variables is put less than $n$ to enable comparison with standard methods.

## Hyperparameter optimal range

Scenario 1 dataset is used to perform the hyperparameter optimization for identifying the optimal ranges of $q$, $Q_i$ and $R_f$. Different hyperparameter combinations are tried with HDSI_AL technique on the training dataset. The hyperparameter combination with best mean predictive performance on the five-fold cross-validated dataset is selected. The best hyperparameters predictive performance is tested on test data. Root mean square error (*RMSE*) is used to measure predictive performance. Table 2 shows the results obtained from 10 trials.

Further, the hyperparameter optimization process is repeated with HDSI_R technique for the same dataset. The hyperparameter range identified for each parameter is within two standard deviation range identified with HDSI_AL technique. Additionally, the RMSE performance obtained from the two techniques is similar. The search region for optimal values of

**Table 2. Optimal region for hyperparameters.**

| HDSI Technique | Hyperparameter (trials = 10) | | | *RMSE* |
|---|---|---|---|---|
|  | *q* | *Q_i* | *R_f* |  |
|  | *Mean (±2SD) [Min, Max]* | *Mean (±2SD) [Min, Max]* | *Mean (±2SD) [Min, Max]* | *Mean (95% CI)* |
| HDSI_AL | 12 (±14) [5,20] | 7.22 (±11.01) [0.73,13.96] | 0.98 (±2.16) [-0.90,2.13] | 0.23(0.19–0.27) |
| HDSI_R | 15 (±4) [9,18] | 6.13 (±6.52) [3.08,15.25] | 0.98 (±1.42) [-0.36,1.74] | 0.25(0.18–0.31) |

**Table 3. HDSI performance in different bootstraps.**

| L | Δ | Bootstraps | Performance (Trials = 10) | | | | | |
|---|---|---|---|---|---|---|---|---|
| | | | HDSI_AL | | | HDSI_R | | |
| | | | Selected Features | | RMSE (95% CI) | Selected Features | | RMSE (95% CI) |
| | | | *Marginal (Range)* | *Interactions (Range)* | | *Marginal (Range)* | *Interactions (Range)* | |
| 1 | 2.8 | 14 | 6 (3–11) | 4 (1–8) | 0.23 (0.19–0.27) | 18 (10–24) | 22 (8–42) | 0.27 (0.25–0.30) |
| 5 | 1.3 | 40 | 3 (3–4) | 1 (1–2) | 0.23 (0.19–0.27) | 10 (5–13) | 8 (3–12) | 0.23 (0.19–0.27) |
| 13 | 0.8 | 80 | 3 (3–4) | 1 (1–2) | 0.23 (0.19–0.27) | 6 (3–7) | 4 (1–5) | 0.23 (0.19–0.27) |
| 32 | 0.5 | 186 | 3 (3–3) | 1 (1–1) | 0.23 (0.19–0.27) | 4 (3–5) | 2 (1–3) | 0.23 (0.19–0.27) |
| 200 | 0.2 | 1006 | 3 (3–3) | 1 (1–1) | 0.23 (0.19–0.27) | 3 (3–4) | 1 (1–2) | 0.23 (0.19–0.27) |

hyperparameters $q$, $Q_i$ and $R_f$ is [2,26], (0, 18.24] and [-1.18,3.14] respectively. The optimal region of $q$ is truncated and depends upon feature space as $q$ cannot take values less than 2 or more than $p$. Similarly, the minimal value of $Q_i$ is truncated at zero as quantile intervals are not negative.

## Bootstraps analysis

Scenario 1 dataset is used to showcase the importance of bootstraps in the model performance. The performance of HDSI_AL and HDSI_R is compared for different effect size and consequently, different number of bootstraps as shown in Table 3. The mean values of hyperparameters $q$, $Q_i$ and $R_f$ optimal region are used for analysis. *RMSE* is used to measure predictive performance. The results from 10 trials suggest that increase in the number of bootstraps reduces the selection of noise variables and improves predictive performance. Effect size, $\Delta$ = 0.5, which corresponds to 186 bootstraps, can eliminate all noise features and only select target features using HDSI_AL technique. In the case of HDSI_R, $\Delta$ = 0.5 can eliminate almost all noise features and only select target features. Nevertheless, an increase in bootstraps helps in reducing the noise features selection. Additionally, results suggest that HDSI could provide good model performance even when globally optimal hyperparameters are not used.

## HDSI comparison with standard methods

The performance of HDSI_L, HDSI_AL and HDSI_R are compared with various standard methods, namely random LASSO, LASSO, adaptive LASSO, group LASSO and regression. The large effect size is considered for the study. The simulation studies are performed in R. Random LASSO is performed by modifying the existing algorithm available at GitHub (https://github.com/samskhan/KSULasso/tree/master/R). The code was modified to allow the algorithm to take different $q$ values and provide the intercept term. Further, the original code was unable to run if the importance score of features after the procedure I is zero, so negligible value ($1/(px10^6)$) is assigned to it. The number of bootstrap samples used for the random LASSO is 200, as suggested by [20] in their paper. The R package *glmnet* is used to perform LASSO and adaptive LASSO [29]. Ridge regression precedes adaptive LASSO (except, when adaptive LASSO is used in random LASSO) for obtaining weights for adaptive LASSO as suggested by [26]. The R package *glinternet* [30] is used to perform group LASSO as it considers interaction terms [31]. The R package MASS [32] is used to perform forward subset selection.

The performance of different methods is evaluated based on multiple criteria. The first criterion is the ability of a method to select true features and reject noise features. Accordingly, the number of target and noise features selected by a method is calculated. The second

criterion is the prediction performance of a method. Root mean square error ($RMSE$) and $R^2$ between the estimated outcome and actual outcome are used as evaluation parameters.

Table 4 shows that among the standard methods, LASSO, Adaptive LASSO and random LASSO showed similar performance in terms of feature selection. They had successfully identified the marginal variables with non-zero coefficients. Similar results were reported by [20] in their paper. Further, LASSO, adaptive LASSO and random LASSO did not select any non-zero marginal coefficients in all the scenarios. These methods outperformed the regression and group LASSO, which consistently selected the noise variables. However, other than Group LASSO, no other standard method was able to identify the interaction variables. HDSI with different selection techniques, i.e. HDSI_L, HDSI_AL and HDSI_R, outperformed the standard methods as they consistently selected the target variables while rejecting almost all noise variables. Therefore, it seems that HDSI might be able to leverage the performance of existing statistical feature selection methods.

Table 5 shows that outcome prediction performance of HDSI is better than the standard methods in all the scenarios. Thus, HDSI seems to be a better option as compared to standard methods in models with interaction terms having non-zero coefficients. Among the standard

**Table 4. Feature selection performance of different approaches in simulated scenarios.**

| Scenario | Performance Parameter (Number of Features Selected) | Standard | | | | | HDSI | | |
|---|---|---|---|---|---|---|---|---|---|
| | | LASSO | Adaptive LASSO | Group LASSO | Random LASSO | Regression | HDSI_L | HDSI_AL | HDSI_R |
| 1 | Marginal ($p = 25$) | 3 | 3 | 25 | 4 | 3 | 4 | 3 | **7** |
| | Target ($s = 3$) | 3 | 3 | 3 | 3 | 3 | 3 | 3 | 3 |
| | Noise ($s = 22$) | 0 | 0 | 22 | 1 | 0 | 1 | 0 | 4 |
| | Interaction ($s = 300$) | 0 | 0 | 78 | 0 | 0 | 2 | 1 | **5** |
| | Target ($s = 1$) | 0 | 0 | 1 | 0 | 0 | 1 | 1 | 1 |
| | Noise ($s = 299$) | 0 | 0 | 77 | 0 | 0 | 1 | 0 | 4 |
| | Total Feature Selection ($s = 4$) | 3 | 3 | 103 | 4 | 3 | 6 | 4 | **12** |
| 2 | Marginal ($p = 50$) | 3 | 3 | 50 | 3 | 5 | 3 | 3 | 7 |
| | Target ($s = 3$) | 3 | 3 | 3 | 3 | 3 | 3 | 3 | 3 |
| | Noise ($s = 47$) | 0 | 0 | 47 | 0 | 2 | 0 | 0 | 4 |
| | Interaction ($s = 1225$) | 0 | 0 | 223 | 0 | 0 | 1 | 1 | 6 |
| | Target ($s = 1$) | 0 | 0 | 1 | 0 | 0 | 1 | 1 | 1 |
| | Noise ($s = 1224$) | 0 | 0 | 222 | 0 | 0 | 0 | 0 | 5 |
| | Total Feature Selection ($s = 4$) | 3 | 3 | 273 | 3 | 5 | 4 | 4 | 13 |
| 3 | Marginal ($p = 100$) | 3 | 3 | 98 | 3 | 3 | 3 | 3 | 6 |
| | Target ($s = 3$) | 3 | 3 | 3 | 3 | 3 | 3 | 3 | 3 |
| | Noise ($s = 97$) | 0 | 0 | 95 | 0 | 0 | 0 | 0 | 3 |
| | Interaction ($s = 4950$) | 0 | 0 | 263 | 0 | 0 | 1 | 1 | 6 |
| | Target ($s = 1$) | 0 | 0 | 1 | 0 | 0 | 1 | 1 | 1 |
| | Noise ($s = 4949$) | 0 | 0 | 262 | 0 | 0 | 0 | 0 | 5 |
| | Total Feature Selection ($s = 4$) | 3 | 3 | 361 | 3 | 3 | 4 | 4 | 12 |
| 4 | Marginal ($p = 50$) | 1 | 1 | 50 | 1 | 1 | 3 | 3 | 24 |
| | Target ($s = 3$) | 1 | 1 | 3 | 1 | 1 | 3 | 3 | 3 |
| | Noise ($s = 47$) | 0 | 0 | 47 | 0 | 0 | 0 | 0 | 21 |
| | Interaction ($s = 1225$) | 0 | 0 | 281 | 0 | 0 | 1 | 1 | 19 |
| | Target ($s = 1$) | 0 | 0 | 1 | 0 | 0 | 1 | 1 | 1 |
| | Noise ($s = 1224$) | 0 | 0 | 280 | 0 | 0 | 0 | 0 | 18 |
| | Total Feature Selection ($s = 4$) | 1 | 1 | 331 | 1 | 1 | 4 | 4 | 43 |

**Table 5. Outcome prediction performance of different approaches in simulated scenarios.**

| Scenario | Performance Parameter (Outcome prediction) | Standard | | | | | HDSI | | |
|---|---|---|---|---|---|---|---|---|---|
| | | *LASSO* | *Adaptive LASSO* | *Group LASSO* | *Random LASSO* | *Regression* | *HDSI_L* | *HDSI_AL* | *HDSI_R* |
| 1 | *Test Dataset* | | | | | | | | |
| | RMSE | 0.47 | 0.47 | 0.27 | 0.51 | 0.43 | **0.26** | **0.26** | **0.26** |
| | $R^2$ | 0.72 | 0.71 | 0.90 | 0.71 | 0.72 | **0.90** | **0.90** | **0.90** |
| 2 | *Test Dataset* | | | | | | | | |
| | RMSE | 0.41 | 0.40 | 0.28 | 0.50 | 0.42 | **0.25** | **0.25** | **0.25** |
| | $R^2$ | 0.76 | 0.76 | 0.87 | 0.75 | 0.73 | **0.89** | **0.89** | **0.89** |
| 3 | *Test Dataset* | | | | | | | | |
| | RMSE | 0.42 | 0.42 | 0.27 | 0.61 | 0.40 | **0.26** | **0.26** | **0.26** |
| | $R^2$ | 0.75 | 0.75 | 0.88 | 0.72 | 0.74 | **0.89** | **0.89** | **0.89** |
| 4 | *Test Dataset* | | | | | | | | |
| | RMSE | 0.41 | 0.40 | 0.29 | 0.48 | 0.42 | **0.25** | **0.25** | 0.26 |
| | $R^2$ | 0.57 | 0.57 | 0.74 | 0.57 | 0.52 | **0.80** | **0.80** | 0.77 |

methods, only group LASSO performance can detect interaction terms and outperformed other standard methods. The performance of HDSI_L, HDSI_AL and HDSI_R in test dataset is coincidentally is identical. Further, HDSI could enable in expanding the modeling functionality of basic statistical approaches like simple linear regression to high dimensional settings if the number of target features is less than *n*. However, the main limitation is that HDSI process is computationally intensive as compared to standard methods.

## Real data studies

We implement the HDSI methods and compare them with other methods on five real-world datasets. The features in these studies can be textual, continuous or categorical with many missing values. For simplicity, the study only uses continuous features and remove features with a large number of missing values.

Dataset I is Community Health Status Indicators (CHSI) dataset (available at https://healthdata.gov/dataset/community-health-status-indicators-chsi-combat-obesity-heart-disease-and-cancer) which contains USA county-level data on various demographics and health parameters to help in making informed decisions in combating obesity, heart disease and cancer. The dataset contains data on 578 features for 3141 US counties. The final dataset has a sample size and feature size of 1156 and 55, respectively.

Dataset II and IV are National Social Life, health and Aging Project (NSHAP) datasets for Wave 3 (2015–2016) (available at https://www.icpsr.umich.edu/icpsrweb/NACDA/studies/36873) and Wave 1 (2005–2006) (available at https://www.icpsr.umich.edu/icpsrweb/NACDA/studies/20541), respectively. The datasets contain data of USA population related to *health*, *social life and well being of older Americans*. The Dataset II contains data on 1470 features for 4377 residents. The final dataset of Dataset II has a sample size and feature size of 1292 and 19, respectively. The Dataset IV contains data on 820 features for 3005 residents. The final dataset of Dataset IV has a sample size and feature size of 1511 and 27, respectively.

Dataset III is Study of Women's Health Across the Nation (SWAN), 2006–2008 dataset (available at https://www.icpsr.umich.edu/icpsrweb/NACDA/studies/32961) which contains multi-site data for middle-aged women in USA on various *physical*, *biological*, *psychological and social* parameters. The dataset contains data on 887 features for 2245 respondents. The final dataset has a sample size and feature size of 1571 and 32, respectively.

Dataset V is Hawaii Aging with HIV Cardiovascular Study dataset (available at https://www.icpsrweb.umich.edu/icpsrweb/NAHDAP/studies/36389) which focus on determining the atherosclerosis development in HIV positive adults with age 40 and over residing in the Hawaii state, USA. The dataset has 248 features related to demographics and health indicators for a sample size of 110. The final dataset has a sample size and feature size of 104 and 21, respectively.

Table 6 provides a detailed summary of the five datasets. Each of the datasets is split into a training dataset ($n_{tr}$) and a test dataset ($n_{te}$). A large effect size is considered for estimating $B$ for reduced computation time. Different methods are compared based on their prediction performance in the test dataset repeated over 30 trials.

Tables 7 and 8 summarise the results of feature selection methods. Firstly, the HDSI methods have performed similar or better than the standard methods, which indicates that the HDSI methods have the potential to compete with existing methods for feature selection task. The RMSE of HDSI is less than or at par with the standard methods for all datasets. In terms of R-squared metric, the performance of proposed and standard methods is similar. Secondly, the HDSI methods have consistently identified the interaction terms. Among the standard methods, group LASSO has also identified the interactions terms, but a high RMSE and many interaction terms indicate overfitting. Thirdly, the performance of different techniques in HDSI varied with the datasets. The similar variation in performance of standard methods based on datasets is observed. HDSI gives a robust performance with different datasets while accommodating the variable performance of specific techniques.

## Conclusion and discussion

An innovative method, HDSI, is proposed to perform variable selection, including the interaction terms in high dimensional settings. The method is inspired by the random forest method provided by [16]. HDSI randomly samples both data and features, along with the incorporation of interaction terms. The method offers the flexibility of generating predictive models using existing modeling techniques available in the literature. The pooling of predictive models developed from different samples addresses the many limitations of shrinkage methods like LASSO and subset selection methods like forward selection. Firstly, it reduces the sample size restriction in feature selection. The current methodology segments the high dimensional feature space to low dimensional feature space to enable the application of classical statistical approaches on the high dimension feature set. Hence, the individual model is restricted by the number of main effects and interaction effects it can accommodate. However, HDSI as a whole is not restricted, since it pools results from multiple restricted models. Secondly, it enables more efficient selection and estimation of interaction terms from existing statistical modeling techniques like LASSO and ordinary least square regression. HDSI does not change the existing methods. Instead, it changes the ecosystem in which the standard methods

**Table 6. Summary of the real datasets.**

| Dataset | Marginal Features ($p$) | Outcome feature | Sample size | | |
|---|---|---|---|---|---|
| | | | Total ($n$) | Train ($n_{tr}$) | Test ($n_{te}$) |
| *Dataset I* | 55 | Percentage of unhealthy days | 1156 | 925 | 231 |
| *Dataset II* | 19 | Height | 1292 | 1034 | 258 |
| *Dataset III* | 32 | Body Mass Index | 1571 | 1257 | 314 |
| *Dataset IV* | 26 | Height | 1511 | 1209 | 302 |
| *Dataset V* | 21 | Framingham Risk Score | 104 | 84 | 20 |

**Table 7. Feature selection performance of different methods on the real datasets.**

| Methods | | Dataset | | | | |
|---|---|---|---|---|---|---|
| | | I | II | III | IV | V |
| | | *Marginal Features (μ (Range))* | | | | |
| Standard | LASSO | 11 (5–30) | 9 (8–11) | 8 (7–10) | 10 (7–18) | 6 (3–10) |
| | Adaptive LASSO | 14 (6–31) | 11 (10–13) | 10 (8–12) | 12 (8–17) | 7 (4–11) |
| | Group LASSO | 53 (50–55) | 19 (19–19) | 7 (7–8) | 26 (25–26) | 20 (19–20) |
| | Random LASSO | 25 (19–31) | 17 (14–19) | 12 (8–14) | 17 (14–21) | 1 (0–2) |
| | Regression | 8 (6–13) | 7 (6–10) | 8 (7–9) | 8 (6–11) | 5 (3–6) |
| HDSI | HDSI_L | 9 (6–15) | 2 (0–2) | 7 (5–9) | 11 (8–14) | **1 (1–2)** |
| | HDSI_AL | **8 (0–13)** | **1 (0–2)** | **6 (0–9)** | **10 (8–14)** | **1 (1–1)** |
| | HDSI_R | 32 (24–38) | 12 (9–14) | 18 (12–24) | 24 (19–26) | 6 (2–12) |
| | | *Interaction Features (μ (Range))* | | | | |
| Standard | LASSO | 0 (0–0) | 0 (0–0) | 0 (0–0) | 0 (0–0) | 0 (0–0) |
| | Adaptive LASSO | 0 (0–0) | 0 (0–0) | 0 (0–0) | 0 (0–0) | 0 (0–0) |
| | Group LASSO | 255 (232–270) | 89 (76–99) | 7 (6–8) | 156 (147–166) | 58 (53–64) |
| | Random LASSO | 0 (0–0) | 0 (0–0) | 0 (0–0) | 0 (0–0) | 0 (0–0) |
| | Regression | 0 (0–0) | 0 (0–0) | 0 (0–0) | 0 (0–0) | 0 (0–0) |
| HDSI | HDSI_L | **3 (1–7)** | **1 (0–1)** | 3 (1–6) | **3 (1–4)** | 0 (0–0) |
| | HDSI_AL | **3 (0–6)** | **1 (0–1)** | **2 (0–5)** | **3 (2–5)** | 0 (0–0) |
| | HDSI_R | 31 (18–44) | 14 (9–17) | 17 (11–22) | 48 (29–61) | **4 (1–8)** |

operate, which enables them to operate on high dimensional data. Thirdly, it is less susceptible to multicollinearity issue. The sampling of feature space separates multicollinear features into different samples. Hence, multicollinear features could be more efficiently selected or removed from the model.

**Table 8. RMSE performance of different methods on the real datasets for test data.**

| Methods | | Dataset | | | | |
|---|---|---|---|---|---|---|
| | | I | II | III | IV | V |
| | | *RMSE (μ (95% CI))* | | | | |
| Standard | LASSO | 0.92 (0.9–0.93) | 3.85 (3.75–3.94) | 0.51 (0.5–0.53) | 3.57 (3.5–3.64) | 0.06 (0.06–0.06) |
| | Adaptive LASSO | 0.91 (0.9–0.93) | 3.83 (3.74–3.92) | 0.51 (0.49–0.53) | 3.57 (3.51–3.64) | 0.06 (0.06–0.07) |
| | Group LASSO | 0.95 (0.93–0.96) | 3.65 (3.56–3.74) | 0.21 (0.2–0.22) | 5.08 (3.85–6.31) | 0.12 (0.09–0.14) |
| | Random LASSO | 0.98 (0.96–1.00) | 3.93 (3.84–4.02) | 1.08 (1.03–1.12) | 3.89 (3.83–3.96) | 0.07 (0.07–0.07) |
| | Regression | 0.91 (0.89–0.92) | 3.74 (3.65–3.83) | 0.50 (0.48–0.52) | 3.55 (3.49–3.61) | 0.06 (0.06–0.06) |
| HDSI | HDSI_L | 0.91 (0.89–0.92) | 3.77 (3.67–3.87) | 0.19 (0.15–0.24) | **3.45 (3.38–3.51)** | **0.06 (0.06–0.07)** |
| | HDSI_AL | 0.91 (0.90–0.93) | 3.83 (3.73–3.92) | 0.31 (0.23–0.40) | 3.46 (3.4–3.53) | **0.06 (0.06–0.07)** |
| | HDSI_R | **0.90 (0.89–0.92)** | **3.6 (3.5–3.7)** | **0.12 (0.11–0.13)** | 6.77 (4.03–9.52) | 0.07 (0.07–0.07) |
| | | $R^2$ *(μ (95% CI))* | | | | |
| Standard | LASSO | 0.45 (0.44–0.47) | 0.27 (0.25–0.29) | 1 (0.99–1) | 0.35 (0.34–0.37) | 0.35 (0.31–0.4) |
| | Adaptive LASSO | 0.45 (0.44–0.47) | 0.28 (0.26–0.29) | 1 (0.99–1) | 0.36 (0.34–0.37) | 0.35 (0.31–0.4) |
| | Group LASSO | 0.44 (0.42–0.45) | 0.34 (0.32–0.36) | 1 (1–1) | 0.3 (0.25–0.35) | 0.18 (0.13–0.24) |
| | Random LASSO | 0.43 (0.42–0.45) | 0.26 (0.25–0.28) | 0.98 (0.98–0.98) | 0.32 (0.31–0.34) | 0.19 (0.15–0.24) |
| | Regression | 0.46 (0.44–0.48) | 0.31 (0.29–0.32) | 1 (1–1) | 0.36 (0.34–0.37) | 0.38 (0.33–0.43) |
| HDSI | HDSI_L | 0.46 (0.44–0.48) | 0.29 (0.27–0.31) | **1 (1–1)** | **0.39 (0.38–0.41)** | 0.13 (0.1–0.17) |
| | HDSI_AL | 0.46 (0.44–0.47) | 0.27 (0.25–0.29) | **1 (1–1)** | **0.39 (0.37–0.4)** | 0.16 (0.12–0.2) |
| | HDSI_R | **0.47 (0.45–0.49)** | **0.35 (0.34–0.37)** | **1 (1–1)** | 0.23 (0.18–0.28) | **0.2 (0.14–0.25)** |

The simulation studies and real-world studies show that HDSI can outperform existing methods in the feature selection and consequently, the prediction performance. Further, the success of HDSI in real data settings demonstrates its practical relevance. Future research could focus on addressing some of the limitations of the current study. The main objective of the study is to explain and propose a method to handle interaction terms during feature selection in high dimensional settings. Hence, the current study has not extensively tested the HDSI with different types of datasets like temporal datasets, categorical outcomes and time to event outcomes and features like categorical features. Such evaluation of current method could determine the robustness of HDSI in real-world scenarios.

Another limitation is that the study has not tried integrating other types of statistical learning methods like *glasso*, decision trees, support vector machines, artificial neural network in HDSI framework. So, it could be an area of exploration to comprehensively determine the capability of HDSI in enhancing the capability of other techniques.

The HDSI model allows the use of different statistical techniques to build models. However, it may not be able to address all the limitations of any given technique. For instance, LASSO results may not be consistent across the bootstraps [33]. Future research could try to develop HDSI approaches which can deal with consistency limitations of techniques.

## Author Contributions

**Conceptualization:** Rahi Jain, Wei Xu.

**Formal analysis:** Rahi Jain.

**Investigation:** Rahi Jain.

**Methodology:** Rahi Jain, Wei Xu.

**Software:** Rahi Jain.

**Supervision:** Rahi Jain, Wei Xu.

**Validation:** Rahi Jain, Wei Xu.

**Writing – original draft:** Rahi Jain.

**Writing – review & editing:** Rahi Jain, Wei Xu.

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
