## [Decision Letter · Decision Letter 0]

14 Aug 2020

PONE-D-20-13743

HDSI: High Dimensional Selection with Interactions Algorithm on Feature Selection and Testing

PLOS ONE

Dear Dr. Xu,

Thank you for submitting your manuscript to PLOS ONE. After careful consideration, we feel that it has merit but does not fully meet PLOS ONE’s publication criteria as it currently stands. Therefore, we invite you to submit a revised version of the manuscript that addresses the points raised during the revision process. You have two detailed reports, with several suggestions on both the technical descriprion of your method and the presentation of the paper. Also please consider to make available the source code.Please submit your revised manuscript by Sep 28 2020 11:59PM. If you will need more time than this to complete your revisions, please reply to this message or contact the journal office at plosone@plos.org. Please include the following items when submitting your revised manuscript:

We look forward to receiving your revised manuscript.

Kind regards,

Fabio Rapallo, Ph.D.

Academic Editor

PLOS ONE

Journal Requirements:

Reviewers' comments:

Reviewer's Responses to Questions

**Comments to the Author**

1. Is the manuscript technically sound, and do the data support the conclusions?

Reviewer #1: Partly

Reviewer #2: Partly

2. Has the statistical analysis been performed appropriately and rigorously? 

Reviewer #1: N/A

Reviewer #2: Yes

3. Have the authors made all data underlying the findings in their manuscript fully available?

Reviewer #1: Yes

Reviewer #2: Yes

4. Is the manuscript presented in an intelligible fashion and written in standard English?

Reviewer #1: Yes

Reviewer #2: Yes

5. Review Comments to the Author

Reviewer #1: The authors proposed a resampling-based method to facilitate feature selection and prediction for high-dimensional data. Technically, the proposed method is an ensemble method and more like a framework, which can be flexible to incorporate any machine learning methods with feature selection including lasso, adaptive lasso and elastic net. Application efficiency should be the key point for this method. However, the paper lacks extensive evaluation and the method itself needs some parameters which are hard or heuristic to determine in real application. More efforts are needed to demonstrate the effectiveness in simulations and real applications. Below are my itemized comments.

Major:

1: In table 3 and table 6, the result of random lasso is far worse than other methods including regular lasso. Random lasso uses two-step bootstrapping to get better estimation with the sacrifice of computing time. It is hard to believe that the performance of random lasso is so poor. Many issues may be reconsidered. a): In the HDSI algorithm, it generates the interaction effect after subspace sampling. That is, after the selection of q features for a bootstrapped sample, all the possible interaction terms among the q features are created. I wonder whether you conduct this procedure similarly to random lasso. In random lasso, you should also generate interaction terms after subspace sampling q features. b): The algorithm in random lasso is quite straightforward and you may directly write your own code to implement it if there is no available R package. The author of the github page you mention is not the original author of random lasso. I doubt the correctness of this github code especially the author mentioned that the code is unable to run unless modification.

2: The number of features selected in each bootstrapped sample, q, is a crucial parameter. In the simulation and real applications, the author directly fixed it at 5 or 15, which is not an appropriate way. The q must be evaluated by cross validation for each simulated data or real application.

3: I believe the number of bootstrapped datasets is not a crucial parameter and there is no value added to estimate the best number of bootstrapped datasets. In random forest, there will be no overfitting along with the increase of number of trees. If computing power allows, the user should choose an adequate large number of trees for the random forest to be stable. Here for HDSI, the concept is similar, and I suggest there is no need to estimate this parameter. If the authors insist to design an algorithm to estimate the appropriate number of bootstrapped datasets, at least simulation studies are needed to show that this parameter indeed has real impact to the performance.

4: For the feature selection part, authors proposed to use three criteria, quantile, confidence interval and R^2 simultaneously, which induce too many parameters which needs to be specified in real applications. The guidance and extensive evaluation should be given.

Minor:

1: Line 160: boostrapped->bootstrapped

2: Please put i=1….B behind min or max in all the notations like min(Rsq_ij) or max(Rsq_ij).

3: The form between line 147 and 148 looks not very nice. Please reformat and polish this form. Also, the form between line 184 and 185.

Reviewer #2: (Since PLoS One does not allow reviewers to use a compiled PDF file as the only 'comments for the authors', I am pasting the exported text below and also attaching the compiled PDF file for the authors' convenience, because some symbols will be surely rendered incorrectly in text-only format.)

The authors have proposed a new method (HDSI) for handling interaction effects in high- dimensional regression by building on the idea of random lasso but making substantial changes to the implementation. Overall, I think it is a new method that attempts to solve an important problem and illustrates its performance through multiple real data analysis and one simulation study.

In what follows, I will restrict my review to technical details and correctness and presentation. Firstly, I think the presentation is a bit pedestrian and could have been better. In particular,

when writing the algorithmic version of HDSI. Also, the motivation can be improved, the benefits of using the proposed method and the issues with random Lasso should be emphasized more in the introduction.

My comments on the paper are below, not necessarily in the order of importance.

1.Since the main contribution of this paper is proposing a new method for handling high- dimensional regression that relies on many tuning parameters, it would be useful to supply the codes for implementing the proposed method in a simulated or real data-set, either through a public repository such as github or an R package. This will also ensure reproducibility and greater use of the method by the community at large.

2.Lasso was proposed in 1996 and since then there have been many, many variants of the Lasso, including extensions for more complicated structures or non-convex penalties or more recent debiased Lasso or LAVA. It seems some discussion of the literature on sparsity is warranted. See Tibshirani [2014].

3.The main motivation for HDSI (or random lasso) is that Lasso can not select more than n fea- tures in large p, small n problems, which is resolved by methods like Elastic net. This brings a natural question, why not compare with elastic net as well, or more advanced regularization methods like de-biased Lasso?

4.Page 13: q2 is a bad notation. Maybe make this q(2)?

5.page 13, line 152: what is meant by ‘little flexibility in coefficient estimation’? Can you elaborate or add a reference?

6.page 14, line 174: same as before? Either explain why random Lasso under-estimates or provide a reference. As a matter of fact, Lasso has a non-vanishing bias in the tails [Carvalho et al., 2009] but it’s not always ‘under’-estimation.

7.Algorithm 2: How do you make sure that the 2q is less than n? It seems this poses a re- striction on how many main effects and interactions can be considered in the model and the simulation study conducted has the problem set to avoid situations where the size of model with interaction terms is larger than the sample size.

8.Algorithm 2: Too many terms that are explained / defined much later in the text, e.g. L, bj,

Rsqij. Please define them before using them in an algorithm.

9.Also comment on Algo 2: Lasso-based CI’s are not reliable [Chatterjee & Lahiri, 2011].

10.page 17, line 221: if you are calculating bootstrap-based CI’s, why care about Normal distri- bution of βˆj ? In fact, bootstrap works better when the statistics under consideration is either non-Gaussian or non-linear.

11.page 18, line 242: R2 does not follow a Normal distribution, might not even be symmetric.

Why use the µ ± f × σ approach? Also, how do you choose f ? (Please do not use ∗ for denoting a product, if necessary use ×.)

12.As explained before, the simulation study is designed in a way where the number of true non- zero β’s and interaction terms are very small (2-4), and it is not clear how that is against the main motivation (limitation of Lasso to select ≤ n coefficients).

13.Among the methods compared (e.g. Table 2), did the regression or lasso/random lasso have any interaction terms in the model at all? It seems only the group lasso had a provision for including interaction terms. If that’s the case, most of the comparisons are meaningless.

14.On Table 2, n is used to denote the number of non-zero coefficients. Statisticians use n for sample size and I think even this paper has n before to denote sample size. Please consider using a different notation.

15.Finally, a minor point: the separation/dichotomy between Statistics-based and ML-based method (Lasso vs Random Forest) is artificial (at least to this reviewer). One can club them both as supervised learning methods or more generally, statistical learning.

References

CARVALHO, C. M., POLSON, N. G. & SCOTT, J. G. (2009). Handling sparsity via the horseshoe.

Journal of Machine Learning Research W&CP 5, 73–80.

CHATTERJEE, A. & LAHIRI, S. N. (2011). Bootstrapping lasso estimators. Journal of the American Statistical Association 106, 608–625.

TIBSHIRANI, R. J. (2014). In praise of sparsity and convexity. Past, Present, and Future of Statis- tical Science , 497–505.

6. PLOS authors have the option to publish the peer review history of their article (what does this mean?). If published, this will include your full peer review and any attached files.

Reviewer #1: **Yes: **Yujia Li

Reviewer #2: No

---

## [Author Response · Author response to Decision Letter 0]

27 Sep 2020

We would like to thank the reviewers for the valuable suggestions and comments which have significantly improved the content and presentation of the paper. We have addressed the specific comments in the point by point response letter.

---

## [Decision Letter · Decision Letter 1]

2 Nov 2020

PONE-D-20-13743R1

HDSI: High Dimensional Selection with Interactions Algorithm on Feature Selection and Testing

PLOS ONE

Dear Dr. Xu,

Thank you for submitting your manuscript to PLOS ONE. After careful consideration, we feel that it has merit but does not fully meet PLOS ONE’s publication criteria as it currently stands. Therefore, we invite you to submit a revised version of the manuscript that addresses the points raised during the review process.

Both reviewers acknowledge major improvements with respect to the previous version, but some comments still needs the authors' attention. In particular, you should address the comment(s) about the hyperparameters. 

We look forward to receiving your revised manuscript.

Kind regards,

Fabio Rapallo, Ph.D.

Academic Editor

PLOS ONE

Reviewers' comments:

Reviewer's Responses to Questions

**Comments to the Author**

1. If the authors have adequately addressed your comments raised in a previous round of review and you feel that this manuscript is now acceptable for publication, you may indicate that here to bypass the “Comments to the Author” section, enter your conflict of interest statement in the “Confidential to Editor” section, and submit your "Accept" recommendation.

Reviewer #1: (No Response)

Reviewer #2: All comments have been addressed

2. Is the manuscript technically sound, and do the data support the conclusions?

Reviewer #1: Yes

Reviewer #2: Yes

3. Has the statistical analysis been performed appropriately and rigorously? 

Reviewer #1: Yes

Reviewer #2: Yes

4. Have the authors made all data underlying the findings in their manuscript fully available?

Reviewer #1: Yes

Reviewer #2: Yes

5. Is the manuscript presented in an intelligible fashion and written in standard English?

Reviewer #1: No

Reviewer #2: Yes

6. Review Comments to the Author

Reviewer #1: I am appreciative of the expanded comments, but dismayed that the investigators did not take into account the rather substantial and insightful comments from both reviewers.

There are some main examples to this:

1: In section “Hyperparameters”, authors only mentioned the need of optimizing the hyperparameters without a practical guidance. There are three parameters needed to be determined: $q$, $Q_i$ and $R_f$, Line 336-340 mentions the hyperparameters are obtained by five-fold CV and heuristic optimization. Could authors state clearly what heuristic procedure they did? In real data (Line 416), authors mention the three parameters are selected by grid search. I am thinking an exhaustive cross validation of the combination of the three parameters seems overwhelming. I suggest authors can clearly state how they select three parameters in “Hyperparameters” Section.

2: Following 1, I suggest authors do some sensitivity analysis to the hyperparameters, giving a practical region of each hyperparameter (i.e. $Q_i$ between 0.05-0.2 works well). With this, users can quickly pick a suitable parameter sets to explore. Also, it helps to reduce the computational load if users want to do cross validation over combinations of three hyperparameters (the number of combinations can be largely reduced).

3: Authors should spend more time to proofread. For example, line 39, p and n should use mathematical format: $p$ and $n$. Line 228: Min_RSq_j is not professional. Line 395, the space is too large and looks strange. There are many flaws like this and authors should proofread carefully and make the writing more professional.

4: Following 3, I feel that many equations can fit into the text. There are 16 equations numbered and most of them are not the key part of this manuscript and it is better to fit them into the text.

5: For number of bootstrapped datasets ($B$), my suggestion is to do a sensitivity analysis using simulation (i.e. B=20, 50, 100, 200, 500, 1000, 2000). At the beginning (B=20), the result will be unstable, but at certain point, the result will be stable (for example, B=500). This sensitivity analysis will be in complementary with the rule of thumb authors proposed.

Reviewer #2: I think almost all the comments that I made on the earlier version were addressed in the current draft. I am still a little bothered by the fact that the current method is essentially bounded by limitation of Lasso of selecting p <= n variables inside the HDSI algorithm. It seems using other variable selection methods might bypass this limitation. I have just a couple of minor suggestions.

a) Include the answer to Q. 7 in the main manuscript, that is, the segment: "The current methodology segments the high dimensional feature space to low dimensional feature space to enable the application of classical statistical approaches on the high dimension feature set. Hence, the individual model is restricted by the number of main effects and interaction effects that it can accommodate. However, HDSI as a whole is not restricted, since it pools results from multiple restricted models."

b) On HDSI algorithm, change "feature technique" to "feature selection technique".

7. PLOS authors have the option to publish the peer review history of their article (what does this mean?). If published, this will include your full peer review and any attached files.

Reviewer #1: No

Reviewer #2: No

---

## [Author Response · Author response to Decision Letter 1]

11 Dec 2020

We are very grateful to you for your effort in handling our paper entitled “HDSI: High Dimensional Selection with Interactions Algorithm on Feature Selection and Testing” [PONE-D-20-13743R1]. With the kind help from you, the Academic Editor Dr. Fabio Rapallo, and two reviewers, the paper has been carefully revised and all the comments are fully dealt with. We have submitted the revised manuscript in the system, along with this cover letter, the response letter to the two reviewers and a document to track changes in the manuscript. After revising the paper by taking all comments from you, the Academic Editor and two reviewers into account, we think that it might be suitable for consideration for publication in PLOS One.

---

## [Decision Letter · Decision Letter 2]

15 Jan 2021

HDSI: High Dimensional Selection with Interactions Algorithm on Feature Selection and Testing

PONE-D-20-13743R2

Dear Dr. Xu,

We’re pleased to inform you that your manuscript has been judged scientifically suitable for publication and will be formally accepted for publication once it meets all outstanding technical requirements.

Kind regards,

Fabio Rapallo, Ph.D.

Academic Editor

PLOS ONE

Additional Editor Comments (optional):

Reviewers' comments:

Reviewer's Responses to Questions

**Comments to the Author**

1. If the authors have adequately addressed your comments raised in a previous round of review and you feel that this manuscript is now acceptable for publication, you may indicate that here to bypass the “Comments to the Author” section, enter your conflict of interest statement in the “Confidential to Editor” section, and submit your "Accept" recommendation.

Reviewer #1: All comments have been addressed

Reviewer #2: (No Response)

2. Is the manuscript technically sound, and do the data support the conclusions?

Reviewer #1: Yes

Reviewer #2: Yes

3. Has the statistical analysis been performed appropriately and rigorously? 

Reviewer #1: Yes

Reviewer #2: Yes

4. Have the authors made all data underlying the findings in their manuscript fully available?

Reviewer #1: Yes

Reviewer #2: Yes

5. Is the manuscript presented in an intelligible fashion and written in standard English?

Reviewer #1: Yes

Reviewer #2: Yes

6. Review Comments to the Author

Reviewer #1: The authors have now addressed all the questions/concerns. These additions/modifications have improved the manuscript.

Reviewer #2: (No Response)

7. PLOS authors have the option to publish the peer review history of their article (what does this mean?). If published, this will include your full peer review and any attached files.

Reviewer #1: No

Reviewer #2: No

---

## [Editor Report · Acceptance letter]

3 Feb 2021

PONE-D-20-13743R2 

HDSI: High Dimensional Selection with Interactions Algorithm on Feature Selection and Testing 

Dear Dr. Xu:

I'm pleased to inform you that your manuscript has been deemed suitable for publication in PLOS ONE. Congratulations! Your manuscript is now with our production department. 

Kind regards, 

on behalf of

Dr. Fabio Rapallo 

Academic Editor

PLOS ONE